# LiDAR Inertial Odometry Based on Indexed Point and Delayed Removal Strategy in Highly Dynamic Environments

**DOI:** 10.3390/s23115188

**Published:** 2023-05-30

**Authors:** Weizhuang Wu, Wanliang Wang

**Affiliations:** College of Computer Science and Technology, Zhejiang University of Technology, Hangzhou 310023, China

**Keywords:** LiDAR-inertial SLAM, dynamic objects, sliding window, dynamic scenarios

## Abstract

Simultaneous localization and mapping (SLAM) is considered a challenge in environments with many moving objects. This paper proposes a novel LiDAR inertial odometry framework, LiDAR inertial odometry-based on indexed point and delayed removal strategy (ID-LIO) for dynamic scenes, which builds on LiDAR inertial odometry via smoothing and mapping (LIO-SAM). To detect the point clouds on the moving objects, a dynamic point detection method is integrated, which is based on pseudo occupancy along a spatial dimension. Then, we present a dynamic point propagation and removal algorithm based on indexed points to remove more dynamic points on the local map along the temporal dimension and update the status of the point features in keyframes. In the LiDAR odometry module, a delay removal strategy is proposed for historical keyframes, and the sliding window-based optimization includes the LiDAR measurement with dynamic weights to reduce error from dynamic points in keyframes. We perform the experiments both on the public low-dynamic and high-dynamic datasets. The results show that the proposed method greatly increases localization accuracy in high-dynamic environments. Additionally, the absolute trajectory error (ATE) and average RMSE root mean square error (RMSE) of our ID-LIO can be improved by 67% and 85% in the UrbanLoco-CAMarketStreet dataset and UrbanNav-HK-Medium-Urban-1 dataset, respectively, when compared with LIO-SAM.

## 1. Introduction

Mobile robots and autonomous driving both require accurate and reliable state estimation. It is challenging for traditional GNSS/INS to achieve high accuracy localization requirements in indoor and urban environments.

In recent years, some developments have been made in both visual and LiDAR SLAM. Due to the wide field of view and detailed texture information that cameras provide, vision-base slam can estimate a six-degree of freedom (6-DOF) state. However, it is easily affected by lighting shifts and low-texture and repetitive-texture environments [1,2]. Compared to cameras, LiDAR has more direct, accurate and reliable depth information for SLAM systems, allowing for robust state estimations even in challenging environments [3].

### 1.1. Localization and Mapping Based on LiDAR and Inertial

The most widely used SLAM technique in urban areas is a localization and mapping algorithm based on LiDAR [4], because LiDAR can accurately estimate the depth of the point cloud. We can categorize these investigations using representations of LiDAR measurements as a basis. The most widely used and basic method, the iterative closest point approach (ICP) [5], uses LiDAR point clouds directly. Recent studies have proposed a variety of representations, such as grid cells, features of LiDAR point cloud and surface elements. The primary distinction between these algorithms is matching search. In particular, to increase computational efficiency, variations in ICP approaches match associated points using downsampled points and various data structures, such as kd-trees, ikd-trees, incremental voxels, and projection depth maps [6,7,8]. The use of projection geometry for a nearest-neighbor search was proposed in [7,9]. SuMa, a powerful 3D LiDAR-based SLAM system that employs a shader language for parallel processing, was proposed by Behley et al. [9]. Using the input point clouds, they executed a spherical image projection and correlated it, treating these points as the closest points. In addition, this paper indicated that motion drift can be greatly decreased through surface element-based map creation and matching. Deep learning, in recent years, has made significant advancements. Many scholars have developed approaches based on deep learning [10,11] for LiDAR motion estimation using projected LiDAR features.

Zhang et al. [12] proposed LiDAR odometry and mapping (LOAM), which is a LiDAR slam solution based on LiDAR features. Instead of using complete point clouds in ICP, planar and edge point features are matched in LOAM to provide real-time and low-drift state estimation. This consists of the odometry system and the mapping system. The odometry module performs point-to-edge/plane matching at a high frequency, and the mapping module runs at a lower frequency to perform scan-to-map matching, which can obtain more precise pose estimations. During the nearest-neighbor search in the former module, kd-trees are built for each feature set from each LiDAR scan. The mapping module uses these LiDAR features to construct a global map and optimize poses further. In the KITTI odometry benchmark site, LOAM has achieved the best performance [13]. In order to further enhance performance, numerous LiDAR slam frameworks based on LOAM have been proposed. F-LOAM [14] uses a non-iterative, two-stage distortion compensation approach to boost the system’s real-time performance and computational efficiency. Nevertheless, there is still no sub-module for global optimization, such as a loop detection strategy. This will lead to significant cumulative errors over long periods of time in large-scale environments. HDL SLAM [15] is a multi-sensor fusion framework that can combine LiDAR, IMU and GNSS sensors; however, the reliability of the frame-to-frame registration is low because it is based on the NDT algorithm. LeGo-LOAM [16] extracts two types of points in the feature extraction module: ground and non-ground points. This significantly improves feature registration efficiency. Then, the LiDAR mapping module receives its initial value via two-step registration. Moreover, a keyframe selection strategy and loopback detection method are added to the backend for better global localization consistency and real-time performance. However, because of the loose coupling of IMU and LiDAR, running in a wide range of scenarios still leads to large cumulative errors. Liu et al. [17] proposed a feature extraction method and computed descriptors based on deep learning. This method uses two-stage state estimation and experiments in a long-range environment. Ye et al. [18] proposed a tightly coupled LiDAR SLAM algorithm, LIO-Mapping, built on VINS and LOAM. For feature extraction and state estimation, a LiDAR front-end component in LOAM takes the role of the visual front-end component in VINS-Mono. Unfortunately, due to the size of the optimization issue, real-time execution on an unmanned vehicle is challenging. LiLi-OM [19] presents an adaptive keyframe selection strategy that can be used for solid-state LiDAR and conventional LiDAR. In addition, it introduces a method with metric weights for sensor fusion. To further increase processing speed and trajectory accuracy, LIO-SAM [20] represents a tightly coupled LiDAR-inertial odometry system based on the framework of incremental smoothing and mapping (iSAM2) [21]. The factor graph optimization can also incorporate GPS and loop closure factors. An algorithm for loosely coupled adaptive covariance estimation was proposed by Zhang et al. [22]. Better experimental results are achieved compared to the LIO system with a tightly coupled framework. Liu et al. [3] proposed a feature extraction algorithm to extract rod-shaped and planar features, which reduces computational consumption and improves the accuracy of the LIO.

In addition to optimization-based LIO systems, filter-based LIO systems have emerged in recent years. The error state Kalman filter is used in [23] to propose a filter-based method for LiDAR-inertial odometry that estimates the robot’s pose state. FAST-LIO [24] represents a new method to calculate the Kalman gain to accelerate the system. FAST-LIO2 [25] directly uses the original LiDAR points to perform scan-to-map registration and proposes the ikd-Tree structure to reduce the time consumption of map updates compared with the static k-d tree structure. Faster-LIO [26] has an iVox structure to organize voxels through a hash table and uses a LRU cache to realize map point addition and deletion, whose parallel accelerated version can achieve a performance that completely surpasses the performance of ikd-Tree. However, these existing SLAM systems are assumed to perform in a static environment. In fact, when an autonomous system navigates in realistic environments, the spatial structure will become more complex as more moving objects, such as moving people and cars, enter the environment. Therefore, for pose estimation and mapping, the online removal of dynamic objects is essential.

### 1.2. Dynamic Point Removal Approaches in SLAM

In the real world, moving objects such as cars and pedestrians are common. However, the excellent SLAM systems mentioned above are designed assuming a static environment and cannot robustly perform in dynamic scenes. Moving objects must be recognized and eliminated from the LiDAR point clouds in order to provide accurate positions and navigation. The following is a summary of the relevant methods:Model-based approaches: These approaches are based on the simple prior model. For example, the ground is required to be removed first in [27,28]. Ref. [29] is based on the concept that most moving objects in urban scenes will inevitably come into contact with the ground.Voxel map-based approaches: These approaches construct a voxel map and track the emitted ray from LiDAR. When the end point of a LiDAR ray hits a voxel, it is considered to be occupied. Moreover, the LiDAR beam is regarded as traveling across free voxels. The voxel probability in the voxel map can be computed in this way. However, these methods are computationally expensive. Even with engineering acceleration in the latest method [30], processing a large number of 3D points online is still difficult [31]. In addition, these methods need highly accurate localization information, which is a challenge for SLAM. In [32], an offline approach for labeling dynamic points in LiDAR scans based on occupancy maps is introduced, and the labeled results are used as training datasets for deep learning-based LiDAR SLAM methods.Visibility-based approaches: In contrast to building a voxel map, the visibility-based approaches just need to compare the visibility difference rather than maintaining a large voxel map [33,34,35]. Specifically, the observed point should be considered dynamic if the view from the previously observed point blocks out the view from the current point. RF-LIO [36] proposed an adaptive dynamic object rejection method based on removert, which can perform SLAM in real time.Learning-based method: The performance of semantic segmentation and detection methods based on deep learning has significantly improved. Ruchti et al. [37] integrated a neural network and an octree map to estimate the occupancy probability. Point clouds in a grid with a low occupancy probability are considered dynamic points. Chen et al. [38] proposed a fast-moving object segmentation network to divide the LiDAR scan into dynamic and still objects. The network is able to operate even faster than the LiDAR frequency. Wang et al. [39] proposed a 3D neural network, SANet, and added it to LOAM for semantic segmentation of dynamic objects. Jeong et al. [40] proposed 2D LiDAR odometry and mapping based on CNN, which used the fault detection of scan matching in dynamic environments.

### 1.3. New Contribution

This paper mainly aims to improve the accuracy of LIO in dynamic urban environments. The following are the main contributions of this paper:An online and effective dynamic point detection method at the spatial dimension is optimized and integrated. This approach fully utilizes the height information of the ground in the point clouds to detect dynamic points.An indexed point-based dynamic point propagation and removal algorithm is proposed to remove more dynamic points in a local map along the spatial and temporal dimensions and detect dynamic points in historical keyframes.In the LiDAR odometry module, we propose a delayed removal strategy for keyframes. Additionally, a lite slide window method is utilized to optimize the poses from scan-to-map module. We assign dynamic weights to the well-matched LiDAR feature points in the historical keyframes in the sliding window.

## 2. Materials and Methods

An overview of our ID-LIO is shown in Figure 1. Our system includes three main modules. The first module is IMU pre-integration. This is employed to estimate system motion and obtain an initial pose result for point cloud decomposition and LiDAR odometry. Second, the feature extraction module includes de-skewed point clouds and feature extraction. In the de-skewed point clouds part, point clouds from a raw LiDAR scan are de-skewed using the pose from the IMU integration. In the feature extraction part, the roughness of the points is computed in order to extract the planar and edge features.

Last, the key component of our LIO system is the mapping module. To achieve dynamic points in local map removal online before the scan-to-map module, several critical steps are applied, as follows: (i) IMU odometry is used to estimate the initial pose. (ii) The raw LiDAR scan and the matching local map are constructed, respectively, as the pseudo occupancies. (iii) The main moving objects of the local map are detected by comparing the height difference in the pseudo occupancy in the scan and the local map. (iv) Through a comparison of the dynamic observation numbers of map points with the threshold, we can finally remove the primary moving points from the local map. After the historical keyframes are updated, the dynamic points in the previous keyframes can also be labeled. (v) If the current frame is determined to be a keyframe, only the LiDAR scan-to-map residuals with dynamic weight are optimized via the lite slide window optimization method.

### 2.1. IMU Pre-Integration

The raw IMU measurements include acceleration ${\hat{a}}_{B}\left(t\right)$ and angular velocity ${\hat{\omega }}_{B}\left(t\right)$. The measured values are all under the IMU coordinate system B, which is the same as the frame of motion body. The IMU measurement model is expressed as:(1)$${\hat{\omega }}_{B}\left(t\right)=\omega_{B}\left(t\right)+b^{g}\left(t\right)+\eta^{g}\left(t\right)$$
(2)$${\hat{a}}_{B}\left(t\right)=a_{B}\left(t\right)-R_{W}^{B}\left(t\right)g^{W}+b^{a}\left(t\right)+\eta^{a}\left(t\right)$$
where *b* is the bias and η is white noise. The raw, measured values of IMU are affected by them. The gravity vector is $g^{W}={\left[0,0,g\right]}^{T}$ in the world frame W. It only affects the measurement of the accelerometer.

In our work, we assume that i+1 is the current frame and *i* is the previous frame. When ${\hat{\omega }}_{B}^{i+1}$ and ${\hat{a}}_{B}^{i+1}$ come, the initial pose state between frame *i* and frame i+1 can be estimated using the IMU pre-integration approach:(3)$$R_{WB}^{i+1}=R_{WB}^{i}\Delta R_{i,i+1}Exp\left(J_{\Delta R}^{\omega }\delta b_{\omega }^{i}\right)$$
(4)$$v_{WB}^{i+1}=v_{WB}^{i}+g^{W}\Delta t_{i,i+1}+R_{WB}^{i}\left(\Delta v_{ij}+J_{\Delta v_{i,i+1}}^{\omega }\delta b_{\omega }^{i}+J_{\Delta v_{i,i+1}}^{a}\delta b_{a}^{i}\right)$$
(5)$$p_{WB}^{i+1}=p_{WB}^{i}+v_{WB}^{i}\Delta t_{i,i+1}+\frac{1}{2}g^{W}\Delta t_{i,i+1}^{2}+R_{WB}^{i}\left(\Delta p_{ij}+J_{\Delta p_{i,i+1}}^{\omega }\delta b_{\omega }^{i}+J_{\Delta p_{i,i+1}}^{a}\delta b_{a}^{i}\right)$$
where $R_{WB}^{i+1}$ is the attitude rotation, $v_{WB}^{i+1}$ is velocity and $p_{WB}^{i+1}$ is position.

### 2.2. Indexed Point Initialization

In our work, we concentrate on feature point clouds in keyframes. We denote the extracted edge and planar features from a LiDAR scan at time step *t + 1* as $F_{t+1}^{e}$ and $F_{t+1}^{p}$, respectively. All the features extracted at time *t + 1* compose a LiDAR keyframe $F_{t+1}$, where $F_{t+1}=\left\{F_{t+1}^{e},F_{t+1}^{p}\right\}$. A number of keyframes of edge points close to the current *t + 1* moment together constitute the local edge point cloud map $M_{t}^{e}$ corresponding to the current frame, where $M_{t}^{e}=\left\{F_{t-i}^{e},F_{t-i+1}^{e},\ldots ,F_{t-1}^{e},F_{t}^{e}\right\}$. i+1 indicates the number of frames that make up the current local corner point map. Similarly, the local plane point cloud map at *t + 1* moment is represented as $M_{t}^{p}$, where $M_{t}^{p}=\left\{F_{t-i}^{p},F_{t-i+1}^{p},\ldots ,F_{t-1}^{p},F_{t}^{p}\right\}$. $M_{t}^{e}$ and $M_{t}^{p}$ compose a local map $M_{t}$ corresponding to the current keyframe $F_{t+1}$, where $M_{t}=\left\{M_{t}^{e},M_{t}^{p}\right\}$.

In order to better find the point and then update the status of the point, a new type of point is defined. This operation of storing extra information in addition to the 3D position of a point is motivated by directed geometry point [41]. In our work, a point **P** is a vector with size 7:(6)$$P={\left[p_{x},p_{y},p_{z},i_{p},i_{kf},c_{p},n_{do}\right]}^{T}$$
where the first 3×1 sub-vector $p={\left[p_{x},p_{y},p_{z}\right]}^{T}$ denotes the point 3D position, and the second 2×1 vector $i=[i_{p},i_{kf}]$ stores the index information of point **p**. $i_{p}$ records the point index in $i_{kf}$, and $i_{kf}$ indicates which keyframe point $i_{p}$ belongs to the keyframe set. In addition, $c_{p}$ is the category of the point. $n_{do}$ is dynamic observation number (DON) that records the number of times the point was observed as a dynamic point.

We do not initialize every LiDAR scan. When a current LiDAR scan is selected as a keyframe, we initialize the points of this keyframe. This is because the local map is composed of multiple keyframes. The state of feature points in a keyframe is what we are concerned with. The difference between a common point and an indexed point is shown in Figure 2. In Figure 2a, the local map consists of multiple keyframes composed of common points. In Figure 2b, because of the indexed point, we can know from which keyframe the points on the local map come. This part is used in Section 2.4.

### 2.3. Dynamic Point Detection

For dynamic point detection in high-dynamic urban environments and in the process of SLAM, the offline post-processed ERASOR [29] method is optimized and integrated into the SLAM online system. As shown in Figure 3, we can see the overview of the online dynamic point detection method.

#### 2.3.1. Problem Definiton

We first denote the current LiDAR frame and corresponding voxel submap at time step *t + 1* as $F_{t+1}^{B}$ and $M_{t}^{W}$, respectively. The former is a full query LiDAR scan and the latter is a feature map that is composed of some keyframes around $F_{t+1}$, found by the sliding window method. Note that $M_{t}^{W}$ is in the world frame **W**. However, $F_{t+1}^{B}$ is in frame **B**. In the ERAOSR algorithm, the point clouds on the local map need to be transformed to frame **B**, and the offline local map needs to be constructed in advance. This takes substantial computational time because of the large number of point clouds on the local map. In order to improve the real-time performance of the system, we use the same local map as the scan-to-map module, and transform frame $F_{t+1}^{B}$ to the world coordinate system **W** before the moving point detection, to obtain $F_{t+1}^{W}$ and record the current position of the motion robot $p_{t+1}^{W}$ in frame **W** first., This can be formulated as follows:(7)$$F_{t+1}^{W}{=}_{B}^{W}T_{t+1}*F_{t+1}^{B}$$
(8)$$p_{t+1}^{W}={\left[x_{t+1}^{W},y_{t+1}^{W},z_{t+1}^{W}\right]}^{T}$$
where ${}_{W}^{B}T_{t+1}$ is the relative transformation between $F_{t+1}^{B}$ and $M_{t}^{W}$ from IMU odometry; Equation (Equation 7) represents the conversion of all point cloud coordinates in the B to the W.

#### 2.3.2. Pseudo Occupancy-Based Removal Method

First, we divide the LiDAR frame $F_{t+1}^{W}$ and the submap $M_{t}^{W}$ to region-wise pseudo occupancy descriptors (R-POD) $O_{t}^{M}$ and $O_{t+1}^{F}$, respectively, in a process similar to scan context:(9)$$O_{t}^{M}=\underset{i\in \left[r_{N}\right],j\in \left[\theta_{N}\right]}{⋃}O_{\left(i,j\right)}^{M}\;O_{t+1}^{F}=\underset{i\in \left[r_{N}\right],j\in \left[\theta_{N}\right]}{⋃}O_{\left(i,j\right)}^{F}$$
where $r_{N}$ is the ring number and $\theta_{N}$ is the sector number. $O_{\left(i,j\right)}^{F\;or\;M}$ is the (i,j)-th bin of R-POD. Let $\theta =arctan2(y-y_{t+1}^{W},x-x_{t+1}^{W})$. Each $O_{\left(i,j\right)}^{F\;or\;M}$ includes the point clouds that meet the following formulas:(10)$$O_{\left(i,j\right)}^{M}=\left\{p_{m}|p_{m}\in M_{t}^{W},\frac{\left(i-1\right)\cdot D_{max}}{r_{N}}\leq \rho_{m}<\frac{i\cdot D_{max}}{r_{N}},\frac{(j-1)\cdot 2\pi }{\theta_{N}}-\pi \leq \theta_{m}<\frac{j\cdot 2\pi }{\theta_{N}}-\pi \right\}$$
(11)$$O_{\left(i,j\right)}^{F}=\left\{p_{f}|p_{f}\in F_{t+1}^{W},\frac{\left(i-1\right)\cdot D_{max}}{r_{N}}\leq \rho_{f}<\frac{i\cdot D_{max}}{r_{N}},\frac{(j-1)\cdot 2\pi }{\theta_{N}}-\pi \leq \theta_{f}<\frac{j\cdot 2\pi }{\theta_{N}}-\pi \right\}$$
where $\rho_{m}=\sqrt{{\left(x_{m}-x_{t+1}^{W}\right)}^{2}+{\left(y_{m}-y_{t+1}^{W}\right)}^{2}}$; $\rho_{f}=\sqrt{{\left(x_{f}-x_{t+1}^{W}\right)}^{2}+{\left(y_{f}-y_{t+1}^{W}\right)}^{2}}$; $D_{max}=\frac{R_{l}}{2}$ and $R_{l}$ is the maximum detection distance of LiDAR. Then, we calculate the maximum height difference $\Delta h_{\left(i,j\right)}^{M}$ and $\Delta h_{\left(i,j\right)}^{F}$ in each bin. Thereafter, we can obtain the potentially dynamic bins in $O_{t}^{M}$ if:(12)$$\Delta h_{\left(i,j\right)}^{M}\;/\;\Delta h_{\left(i,j\right)}^{F}>0.2$$

Finally, we use region-wise ground plane fitting (R-GPF) [29] to obtain ground points in dynamic bins and the ground plane. The dynamic points are points in dynamic bin, expecting ground points, and all dynamic points in dynamic bins consist of the final dynamic points set $M_{D}^{m}$. Therefore, we can obtain the dynamic set $M_{D}$, where $M_{D}$ is
(13)$$M_{D}=M_{D}^{m}$$

### 2.4. Dynamic Point Propagation and Removal

ERAOSR [29] is an offline method that uses the long sequence local map. However, we need to eliminate dynamic points in the online process of SLAM. We can only obtain the short sequences in the SLAM, which will decrease the performance of dynamic point removal. Therefore, we propose a dynamic point removal method based on spatial and temporal dimensions to improve the removal performance.

In our work, we do not remove dynamic points in a submap when obtaining the dynamic point set $M_{D}$ after the dynamic point detection module, like RF-LIO. We judge whether the feature point is dynamic not only according to the difference between the current LiDAR scan and submap but also via the history difference. We propagate these points to historical keyframes based on the indexed points in Section 2.2. Then, we perform the dynamic point removal. If a feature point is a dynamic feature point, the frequency with which the feature point is judged to be a dynamic feature point will be high. Thus, the dynamic observation numbers (DON) of a point is used to determine whether the point is dynamic in the temporal dimension. The dynamic point propagation and removal process is summarized in Algorithm 1.
**Algorithm 1:** Dynamic Point Propagation and Removal Algorithm
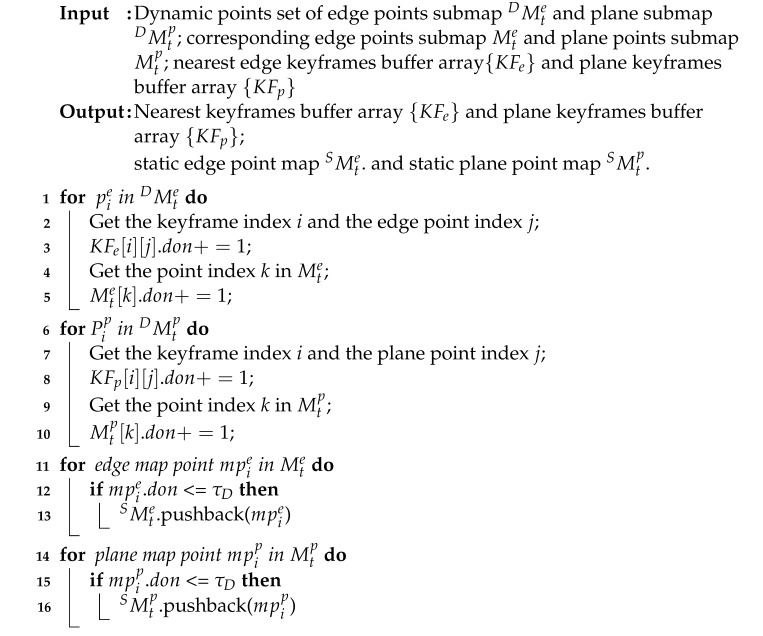


In this algorithm, $\tau_{D}$ is a threshold used to distinguish dynamic feature points and static feature points. If $\tau_{D}$ is small, there will be many static points misclassified as dynamic points. If $\tau_{D}$ is large, we cannot remove dynamic feature points better. In our system, $\tau_{D}$ is set to 3 and 3 is an empirical value; it ensures a balance between the removal of dynamic points and the retention of static points.

### 2.5. LiDAR Odometry

This module includes scan-to-map matching and front-end optimization. To describe this section clearly, we start with the scan-to-map matching module, which is followed by the front-end optimization module.

#### 2.5.1. Feature-Based Scan-to-Map Matching

The sensor motion between two consecutive LiDAR scans is estimated using the front end of SLAM. There are a variety of scan-matching methods, and the fundamental method of this module is the same as LOAM. We compute the roughness of each point in each LiDAR frame. Then, we extract edge and planar features for this new scan. The formula for calculating roughness is as follows:(14)$$roughness=\frac{1}{n.||p_{i}\left|\right|}\left|\right|\sum_{j=1,j\neq i}^{n}\left(p_{j}-p_{i}\right)\left|\right|$$
where $p_{i}$ represents the target point and $p_{j}$ is in the same ring as the LiDAR scan. The edge and planar features from the LiDAR frame t+1 are denoted as $F_{t+1}^{e}$ and $F_{t+1}^{p}$, respectively. Then, the feature points in $F_{t+1}^{e}$ and $F_{t+1}^{p}$ are transformed to *W* and obtain $\left\{{}^{W}F_{t+1}^{e}{,}^{W}F_{t+1}^{p}\right\}$. IMU pre-integration provides the initial transformation. By using the nearest-neighbor search method, we are able to locate its correlating feature points in $\left\{M_{t}^{e},M_{t}^{p}\right\}$. Thus, the following formula can be used to calculate the distance between a target point and its related edge:(15)$$d_{t}^{e}=\frac{\left(p_{t+1}^{ei}-m_{t}^{e1}\right)\times \left(p_{t+1}^{ei}-m_{t}^{e2}\right)}{|m_{t}^{e1}-m_{t}^{e2}|}$$
where $p_{t+1}^{ei}\in^{W}F_{t+1}^{e}$ is an edge feature point and $m_{t}^{e1},m_{t}^{e2}\in M_{t}^{e}$ are two different points on the corresponding edge line. Similarly, the distance from a target point to its associated plane can be calculated as follows:(16)$$d_{t}^{p}=\frac{|{\left(p_{t+1}^{pi}-m_{t}^{p1}\right)}^{T}\left(\left(m_{t}^{p1}-m_{t}^{p2}\right)\times \left(m_{t}^{p1}-m_{t}^{p3}\right)\right)|}{|\left(m_{t}^{p1}-m_{t}^{p2}\right)\times \left(m_{t}^{p1}-m_{t}^{p3}\right)|}$$
where $p_{t+1}^{pi}\in^{W}F_{t+1}^{p}$ is an planar feature point and $m_{t}^{p1},m_{t}^{p2},m_{t}^{p3}\in M_{t}^{p}$ are three different points on the corresponding plane.

At last, we can estimate the pose between the current frame and local map by solving the optimization problem:(17)$$\min_{\Delta T_{t+1}^{t}}r_{L}=\min_{\Delta T_{t+1}^{t}}\left\{\sum_{p_{t+1}^{ei}\in^{W}F_{t+1}^{e}}d_{t}^{e}+\sum_{p_{t+1}^{pi}\in^{W}F_{t+1}^{p}}d_{t}^{p}\right\}$$

#### 2.5.2. Front-End Optimization

Within the scanning-to-mapping part, the local map is used with the dynamic points removed. However, we use a LiDAR scan, in which dynamic points exist. Because, in a single LiDAR frame, there are not enough sparse LiDAR points to use the dynamic point removal method as described above [42], we propose a delayed removal strategy for the keyframe and build a cost function including only LiDAR measurements with a dynamic observation-related weight jointly motivated by VINS-Mono. To obtain more accurate odometry for each keyframe of LiDAR, we optimize the pose states in the lightweight sliding window iteratively.

In our LIO system, the states in the sliding window that need to be optimized are defined as $X^{w}=\left[x_{t-n}^{w},x_{t-n+1}^{w},x_{t-1}^{w},\ldots ,x_{t}^{w}\right]$, where we optimize the poses of the n keyframes before the current moment **t**, and $x_{t-i}^{w}=\left[p_{b_{t-i}}^{W},q_{b_{t-i}}^{W}\right]$. For a sliding window of size n, these states are obtained by minimizing the final cost function, which is described as:(18)$$\min_{X^{w}}\left\{\sum_{k=0}^{n}r_{L}^{e}\left(x_{k}^{w}\right)+\sum_{k=0}^{n}r_{L}^{p}\left(x_{k}^{w}\right)\right\}$$

In this cost function, $r_{L}^{e}\left(x_{k}^{w}\right)$ denotes edge-line geometric terms of LiDAR and $r_{L}^{p}\left(x_{k}^{w}\right)$ means the planar LiDAR error terms. Based on the previous LiDAR term calculation of each scan, the LiDAR term incorporates geometric constraints with dynamic weight from the dynamic point propagation and removal module (see Section 2.4) into the cost function scheme. The term is denoted as:(19)$$r_{L}\left(x_{k}^{w}\right)=\sum_{i}^{m}w_{i}d_{i}^{e}+\sum_{j}^{n}w_{j}d_{j}^{p}$$
and the dynamic observations-related weight is defined as:(20)$$w=\left\{\begin{array}{ccc} 1 & \; & n_{do}<=1\\ \frac{1}{n_{do}} & \; & n_{do}>=2 \end{array}\right.$$
where $d_{i}^{e}$ is the distance from the *i*-th target edge point to its corresponding edge and $d_{j}^{p}$ is the distance from target planar point to its corresponding plane in Section 2.5.1; $w_{i}$ is a dynamic observations-related weight; and $n_{do}^{i}$ is the dynamic observations of the *i*-th feature point. In this paper, we set the value of dynamic weight according to the DON. This indicates that the higher the DON, the less important the corresponding residuals *d* are in the cost function *r*.

## 3. Results

### 3.1. Experimental Setup

We validate our system ID-LIO by conducting a set of experiments on public datasets, and ID-LIO is contrasted with FAST-LIO2, Faster-LIO and LIO-SAM. Compared with the LIO-SAM, the ID-LIO employs the same feature extraction module, loopback detection strategy and optimization method for the back-end using GTSAM. We use evo [43] to evaluate the accuracy. The ablation experiment demonstrates the effect of our proposed methods. The proposed methods are executed on a computer with an Intel i5 CPU and 32G RAM. Ubuntu 18.04 with the robot operating system (ROS) melodic [44] is our computer system. The datasets used for validation include UrbanNav datasets in Hong Kong (UNHK) [45] and UrbanLoco datasets in California (ULCA) [46]. The UrbanNav-HK-Data20190428 is a dataset containing some movable objects that is defined as a low-dynamic dataset. We define the others in UNHK and ULCA as high-dynamic datasets because they contain masses of movable objects. As shown in Table 1, the data information is listed in detail. The sensor suite of the UBHK dataset is shown in Figure 4.

### 3.2. Dynamic Observations Number Analysis

The impact of the appropriate DON value on the removal results of dynamic objects is very critical. Consequently, the purpose of this experiment is to demonstrate that the approach mentioned in Section 2.4 is accurate. Setting the correct DON threshold is important. If the DON value is too low, the static points may be incorrectly regarded as dynamic points. If it is too large, many moving points may be wrongly identified as static points. In the implementation of our algorithm, we set the DON value to 3. This can ensure that static points and dynamic points are correctly distinguished. To further prove the rationality of this threshold value, we chose three dynamic urban datasets. We tested the LIO-SAM using the different thresholds between 1 and 6. DON1 to 6 means the value of DON is from 1 to 6. The root mean square error (RMSE) of the absolute trajectory error (ATE) of them is shown in Table 2, which demonstrates that the fixed threshold 3 can provide more reliable and accurate pose estimation.

ULCA-MarktStreet was chosen for our analysis. Figure 5 shows the trajectories of different DONs with our system in ULCA-MarktStreet in an X–Y plot when the thresholds are 1, 3 and 6. This figure shows the total error for the three distinct thresholds. It can be seen that the trajectory corresponding to DON3 is closer to the ground truth because the DON3 system achieves a balance between dynamic point removal and static point retention.

The box diagram of the deviation is shown in Figure 6, which allows us to have a better view of the magnitude of the deviation. The DON3 system has the smallest error.

As shown in Figure 7, there are two kinds of red point sets in the figures. Red cloud points on the street indicate dynamic objects by moving cars, and the more red points, the better. However, red cloud points at other places indicate static objects that have been incorrectly removed, and the fewer red points, the better. If the threshold is 1, more dynamic points are removed. At the same time, many static points are incorrectly removed. If the threshold is 6, more static points are preserved. However, many dynamic points are also preserved in comparison with DON1. Neither of these two extreme cases applies to dynamic point removal. However, DON3 can achieve better rejection results. It retains most of the static points of the local map while removing more dynamic points.

### 3.3. Comparison of Dynamic Removal Strategies for Local Map

To further demonstrate the efficacy of the proposed removal strategy based on the pseudo occupancy and indexed point in the spatial and temporal dimensions (denoted as LIO-SAM + Spatial + Temporal), we make comparisons with the removal strategy based on the pseudo occupancy in the spatial dimension only (denoted as LIO-SAM + Spatial). The ATEs of the mapping thread are shown in Table 3. Figure 8a shows the dynamic point removal results obtained by these two methods. The red points on the street are dynamic traces by moving cars. Figure 8b identifies part of the dynamic feature points. In contrast, our method based on spatial and temporal dimensions successfully removes most of the dynamic traces. After the trajectories of both systems are aligned with the ground truth, we acquire the error maps for these two approaches, which are illustrated in Figure 9.

### 3.4. Comparison of Delayed Removal Strategy for Keyframe

During the scan-to-map matching, both the dynamic points in the map and the LiDAR scan can result in significant deviation. According to the moving points on the local map, we can obtain the dynamic points in keyframes. When we perform the slide window optimization, the dynamic LiDAR terms will obtain the dynamic weights. Although we cannot remove dynamic points in the current keyframe, they will be removed in the next slide window optimization. Therefore, we conducted this experiment to prove the efficiency of the delayed removal strategy. The ATEs of the mapping thread are shown in Table 4. Figure 10 shows the trajectories of our system with delayed removal strategy and without the strategy. Figure 11 shows the error maps of our system without different removal strategy. Figure 12 shows the removal result. Figure 12a,b shows the real environment with moving cars and people corresponding to the current frame. Figure 12d shows a keyframe with red dynamic points after the delayed removal strategy. The dynamic cars and people are marked with red points. Figure 12c shows a static keyframe after removing the dynamic points of this keyframe.

The goal of this experiment is to demonstrate the effectiveness of the delayed removal method for dynamic points in historical keyframes. We use a dynamic point propagation method to obtain the dynamic points in historical keyframes based on the dynamic local map feature points. Figure 12f shows the dynamic traces on the local map. Figure 12e shows the dynamic points in this keyframe obtained by the dynamic point propagation method; for example, red points marked by yellow rectangles in Figure 12. There are five people walking on the road. These red points are part of the red points in the yellow rectangle in Figure 12f. Similarly, the car in the yellow oval in Figure 12e is part of the red points in the yellow oval in Figure 12f. The error maps of the ID-LIO with delayed removal strategy and without this strategy are shown in Figure 11. We can see that the trajectory in our system with this method is closer to the ground truth.

### 3.5. Results on Low- and High-Dynamic Datasets

We evaluate our ID-LIO system on six datasets with the ground truths from the UrbanNav and UrbanLoco datasets, and we compare the results with LIO-SAM, Fast-LIO, Faster-LIO and the ground truths. For quantitative evaluation, we computed the ATE for these three methods on different datasets using the evaluation tool EVO, and the results are shown in Table 5.

After the trajectories of these four systems are aligned with the ground truth, we acquire the error maps for these approaches. These are illustrated in Figure 13, from which we can see that the error of each position calculated by our system is less than 1.10 m, while the error of Faster-LIO, FAST-LIO and LIO-SAM can be up to 9.81 m, 9.34 m and 7.51 m, respectively.

Finally, we compare the differences between the point cloud maps of FAST-LIO and ID-LIO in Figure 14. We can see that Fast-LIO is the least effective because it does not have back-end loopback detection, and LIO-SAM is unable to detect the closed-loop accurately because there are many dynamic objects in the urban environment, resulting in inaccurate poses obtained by scan2map. Our ID-LIO adopts a dynamic point detection and rejection algorithm and achieves better loopback results compared to LIO-SAM.

### 3.6. Runtime Performance Analysis

Real-time performance is also a critical criterion to consider when evaluating SLAM systems in practice. We record the time consumption of ERASOR, LIO-SAM and ID-LIO on these six datasets. We can see that the runtime of ERASOR is more than 100 ms because of the amount of time spent on map construction and point cloud coordinate conversion. LIO-SAM is a real-time SLAM system that processes each frame in less than 100 ms. ID-LIO is a system with a dynamic point removal module that performs well in real time in five datasets. ULCA-MarktStreet is a highly dynamic and large scale dataset. Therefore, the runtime of ID-LIO is more than 100 ms. The runtime results are shown in Table 6.

## 4. Discussion

In this paper, we proposed an improved and robust LIO system in dynamic environments. A dynamic point removal method based on pseudo occupancy and indexed points is used for a local map before scan-to-map registration. It reduces the influence of dynamic points on feature matching and improves pose accuracy. In addition, we proposed a delayed removal strategy for keyframes, and the optimization based on sliding window incorporates the LiDAR measurement with dynamic weights to reduce error from dynamic points in keyframes, which further improves the pose accuracy. ID-LIO enables the real-time, accuracy and robustness requirements of SLAM. Compared with the offline ERASOR algorithm, our algorithm is 40% faster. In addition, the results of our dynamic object removal algorithm are better than ERASOR. We can remove more dynamic points and preserve more static points, as shown in Figure 9. The ATE average RMSE of our ID-LIO can be improved by 67% and 85% in the UrbanLoco-CAMarketStreet dataset and UrbanNav-HK-Medium-Urban-1 dataset, respectively, when compared with LIO-SAM. The dynamic point removal module enhances the robustness and accuracy of ID-LIO in highly dynamic environments. Compared with FAST-LIO and Faster-LIO, the ATE average RMSE of our ID-LIO can be improved by 88% and 89%, respectively, in the UNHK-TST dataset.

For future work, we note that it is important to improve the real-time performance of the LIO system in highly dynamic and large-scale environments. We can see that in the ULCA-MarktStreet dataset, the runtime of our system is more than 120 ms. This is because we need to construct local maps for scan-to-map registration and pseudo occupancy maps for dynamic point removal repetitively in our system, which is time-consuming. In the future, we will focus on spatial data structure for local map addition and deletion to improve the system’s efficiency. It is also important to note that the LIO system is susceptible to Z-direction drift in vast scenarios. In the future, the ground constraint will be added to reduce drift on the Z-axis.

## Figures and Tables

**Figure 1 sensors-23-05188-f001:**
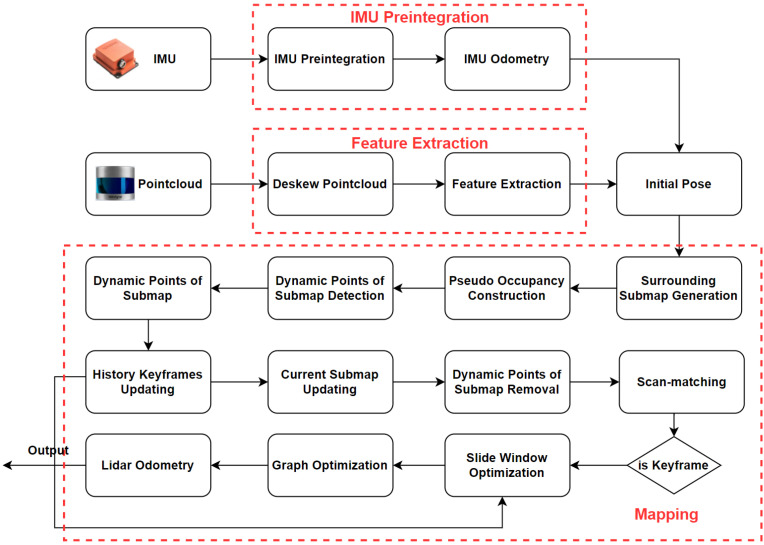
An overall framework of ID-LIO system.

**Figure 2 sensors-23-05188-f002:**
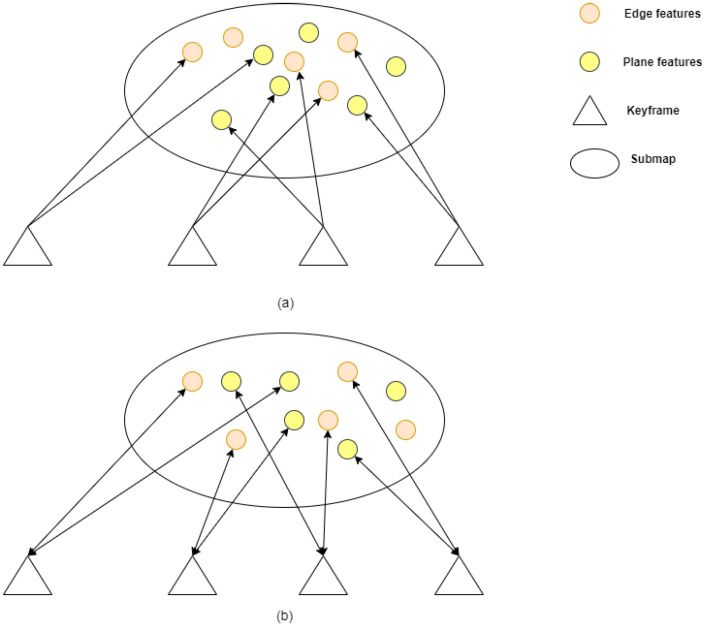
The difference between common points and index points. In (**a**), the local map consists of multiple keyframes composed of common points. In (**b**), the local map consists of multiple keyframes composed of indexed points. In this way, we can know from which keyframe the points on the local map come.

**Figure 3 sensors-23-05188-f003:**
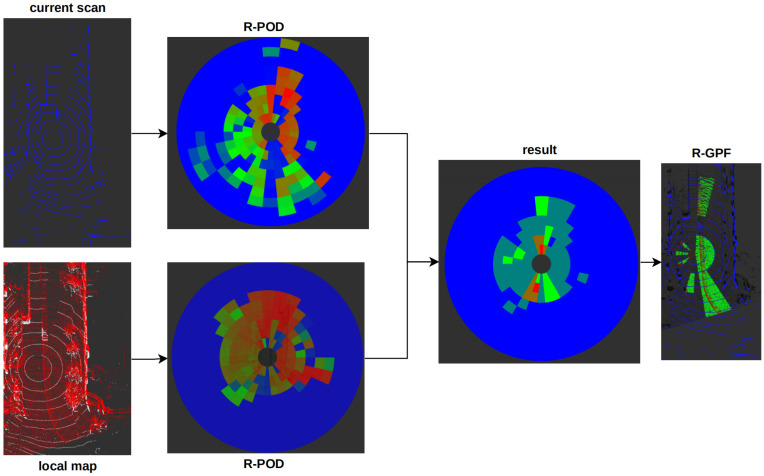
Overview of the online dynamic point detection method. First, we construct R-POD for the current frame and local map, respectively. Then, we compare the height difference in the pseudo occupancy in the scan R-POD and local map R-POD and obtain the dynamic pseudo occupancy of the local map. Last, we use the R-GPF method to obtain the ground plane, and the points above the ground are dynamic points.

**Figure 4 sensors-23-05188-f004:**
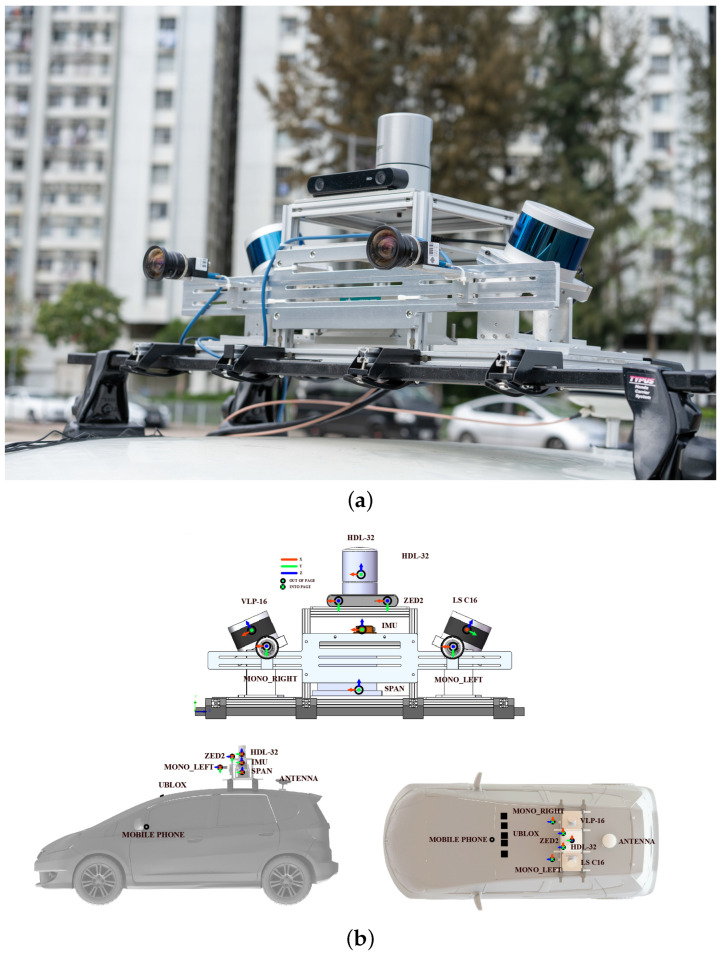
The sensor suite of the UNHK [45] dataset. (**a**) is a view of the sensor kit in the car. (**b**) shows the distributions location of each sensors.

**Figure 5 sensors-23-05188-f005:**
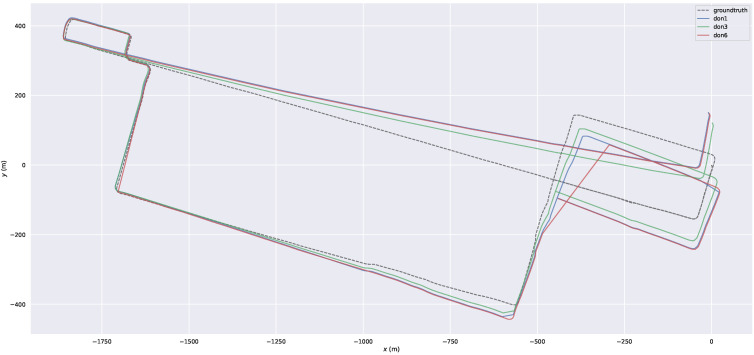
The trajectories of the three DON thresholds on ULCA-MarktStreet. The trajectory of DON3 is the most accurate; it is closer to the ground truth.

**Figure 6 sensors-23-05188-f006:**
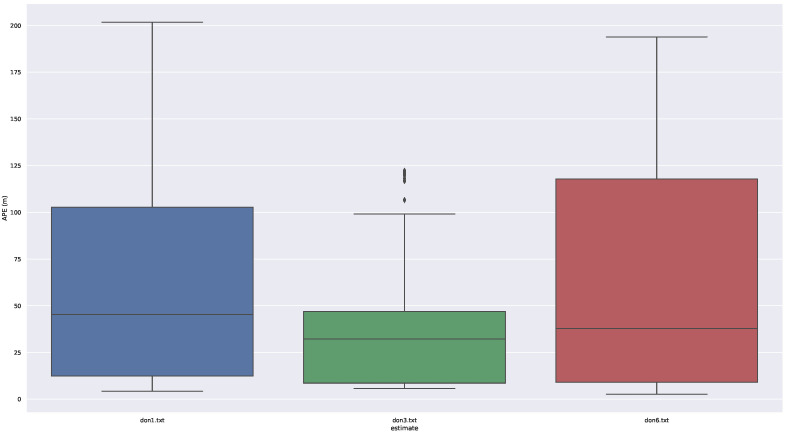
Box diagram of the deviation.

**Figure 7 sensors-23-05188-f007:**
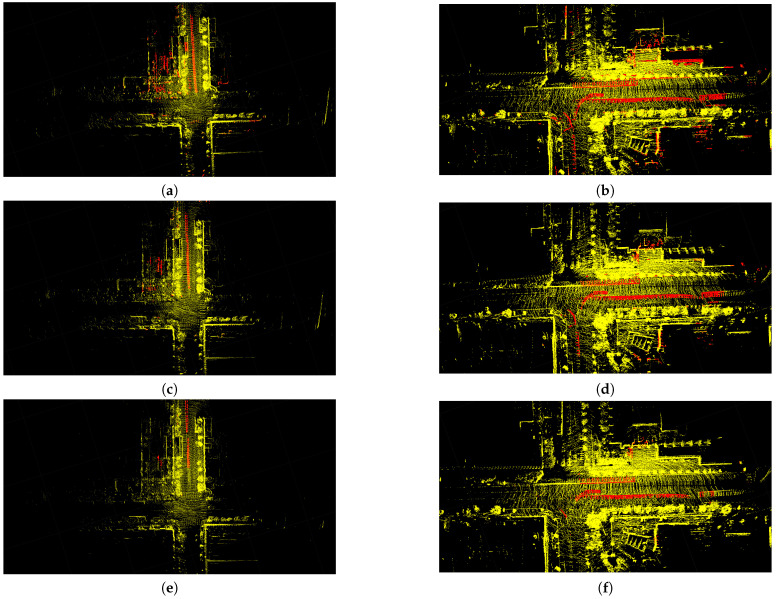
The removal results of local maps corresponding to different DON thresholds. (**a**,**c**,**e**) are the local maps corresponding to different DON thresholds in keyframe 110. (**b**,**d**,**f**) are the local maps corresponding to different DON thresholds in keyframe 310.

**Figure 8 sensors-23-05188-f008:**
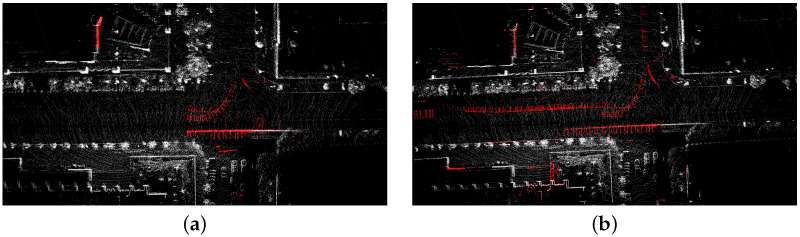
Comparison of dynamic point removal results produced by different removal methods. Dynamic objects are represented by red dots, and it is better if there are more red points. (**a**) Dynamic point removal results by removal method based on spatial dimension. (**b**) Dynamic point removal results by removal method based on spatial and temporal dimension.

**Figure 9 sensors-23-05188-f009:**
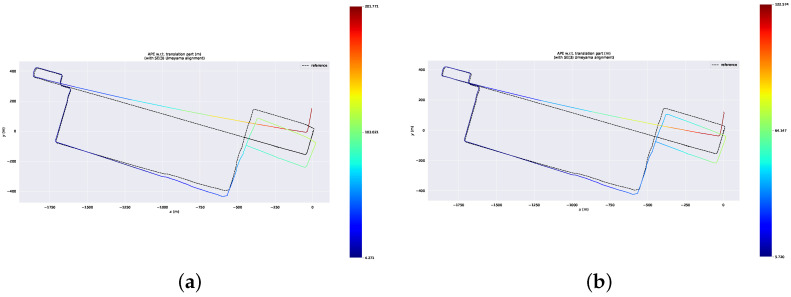
Error map of two dynamic removal methods. (**a**) Error map of removal method based on spatial dimension. (**b**) Error map of removal method based on spatial and temporal dimension.

**Figure 10 sensors-23-05188-f010:**
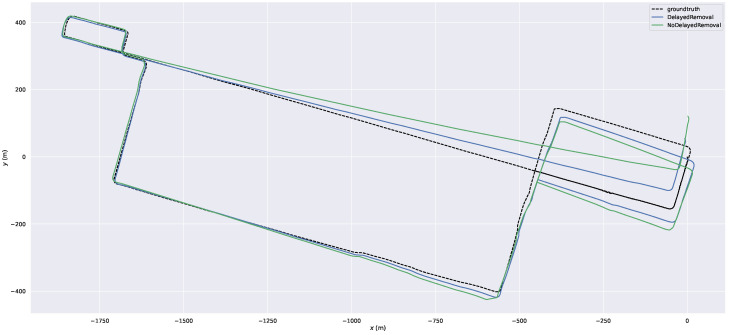
The trajectories of our system with delayed removal strategy and without the strategy.

**Figure 11 sensors-23-05188-f011:**
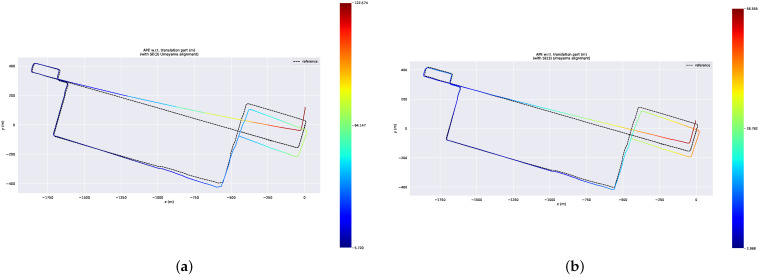
Error map of our system without different removal strategy. (**a**) Error map of our system with delayed removal strategy. (**b**) Error map of our system with the strategy.

**Figure 12 sensors-23-05188-f012:**
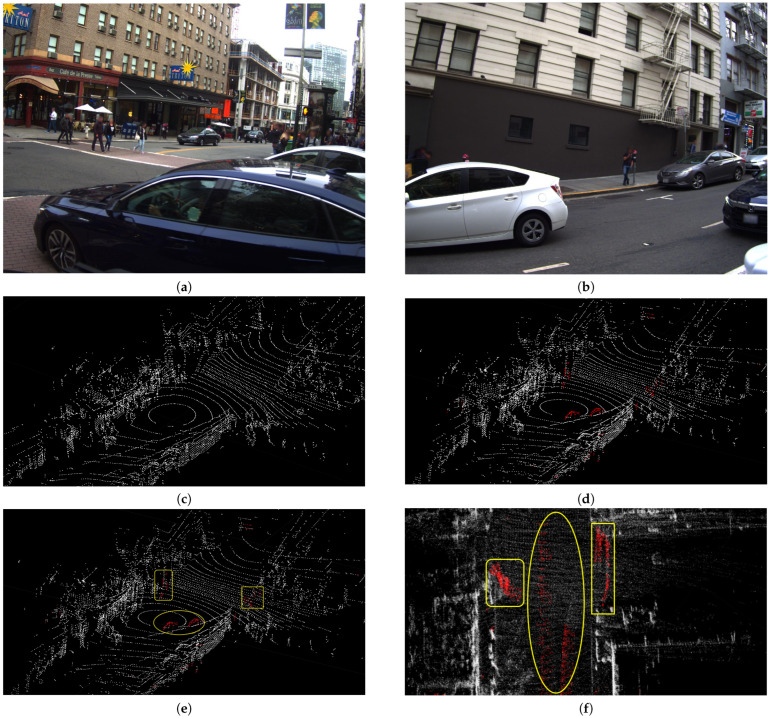
Keyframe with red dynamic points after the delayed removal strategy. (**a**,**b**) show the real environment with moving cars and people corresponding to the current frame. (**c**) shows a static keyframe after removing the dynamic points of this keyframe. (**d**) shows a keyframe with red dynamic points after the delayed removal strategy. (**e**) shows the dynamic points in this keyframe obtained by the dynamic point propagation. (**f**) shows the dynamic traces on the local map.

**Figure 13 sensors-23-05188-f013:**
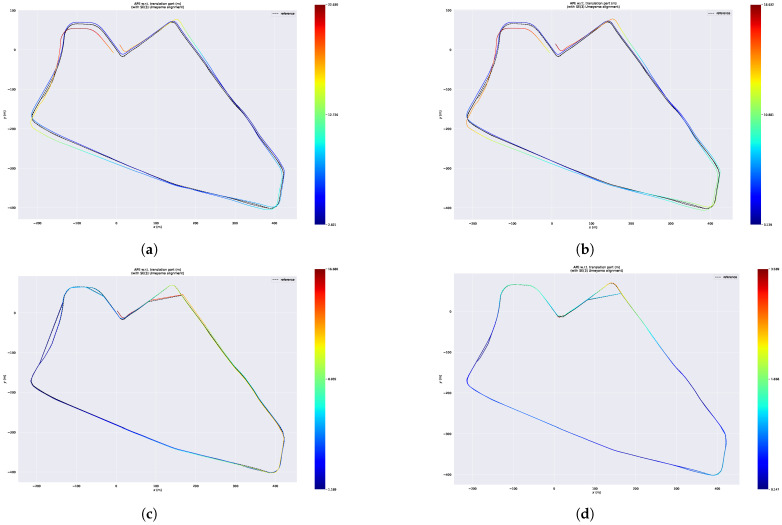
Error maps of four systems on UNHK-TST dataset. (**a**) Error map of Faster-LIO. (**b**) Error map of Fast-LIO. (**c**) Error map of LIO-SAM. (**d**) Error map of ours.

**Figure 14 sensors-23-05188-f014:**
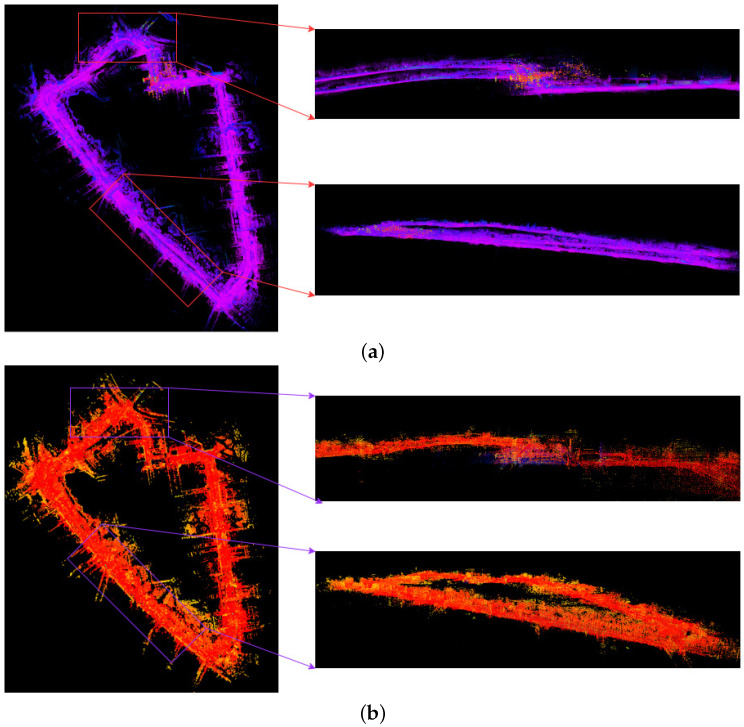
FAST-LIO and ID-LIO mapping results on the UNHK-TST. (**a**) Mapping result of FAST-LIO. (**b**) Mapping result of ID-LIO.

**Table 1 sensors-23-05188-t001:** UrbanNav and UrbanLoco datasets’ details.

Dataset	Trajectory Length (m)	Dynamic Level	Scale Level
UNHK-Data20190428 [45]	1800	Low	Medium
UNHK-TST [45]	3640	High	Medium
UNHK-Mongkok [45]	4860	High	Medium
UNHK-Whampoa [45]	4510	High	Medium
ULCA-MarketStreet [46]	5690	High	Large
ULCA-RussianHill [46]	3570	High	Medium

**Table 2 sensors-23-05188-t002:** RMSE of ATE of different DON thresholds in three datasets.

Sequence/ATE	DON1	DON2	DON3	DON4	DON5	DON6
UNHK-Data20190428 [45]	6.55	6.36	5.96	6.55	6.34	6.43
UNHK-TST [45]	7.51	4.86	4.27	4.70	5.20	7.39
ULCA-MarktStreet [46]	86.69	87.500	49.26	49.98	49.75	86.05

**Table 3 sensors-23-05188-t003:** RMSE of ATE for LIO-SAM with dynamic removal method based on spatial dimension (denoted as LIO-SAM + Spatial) and LIO-SAM with dynamic removal method based on spatial and temporal dimensions (denoted as LIO-SAM + Spatial + Temporal).

Sequence/ATE	LIO-SAM + Spatial	LIO-SAM + Spatial + Temporal
UNHK-Data20190428 [45]	6.55	5.96
UNHK-TST [45]	7.51	4.27
ULCA-MarktStreet [46]	86.69	49.26

**Table 4 sensors-23-05188-t004:** RMSE of ATE for our system without delayed removal strategy (denoted as Ours + No Delayed Removal) compared to ours with the strategy (denoted as Ours + Delayed Removal).

Sequence/ATE	Ours + No Delayed Removal	Ours + Delayed Removal
UNHK-Data20190428 [45]	5.96	5.99
UNHK-TST [45]	4.86	1.06
ULCA-MarktStreet [46]	49.26	28.02

**Table 5 sensors-23-05188-t005:** RMSE of ATE on UNHK and ULCA datasets (m).

Sequence/ATE	Faster-LIO	FAST-LIO	LIO-SAM	Ours
UNHK-Data20190428 [45]	7.53	7.46	6.55	5.96
UNHK-TST [45]	9.81	9.34	7.51	1.06
UNHK-Mongkok [45]	10.45	10.65	8.89	3.45
UNHK-Whampoa [45]	5.13	5.38	3.32	0.85
ULCA-MarktStreet [46]	-	-	86.69	28.02
ULCA-RussianHill [46]	100.56	110.37	60.35	15.34

**Table 6 sensors-23-05188-t006:** Average runtime of three systems for processing a scan (ms).

Sequence	ERAOSR	LIO-SAM	ID-LIO
UNHK-Data20190428 [45]	130.5	50.4	89.5
UNHK-TST [45]	135.8	54.8	95.5
UNHK-Mongkok [45]	138.4	68.6	98.9
UNHK-Whampoa [45]	135.1	55.3	97.8
ULCA-MarktStreet [46]	140.6	80.5	120.6
ULCA-RussianHill [46]	134.5	65.3	99.5

## Data Availability

Not applicable.

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
