# Peer review of "LiDAR Inertial Odometry Based on Indexed Point and Delayed Removal Strategy in Highly Dynamic Environments"

_sensors, 2023, doi:10.3390/s23115188_

Round 1

Reviewer 1 Report

Comments in tha attached pdf file

Author Response

My replay is in a Word file.  We would like to thank the referee again for taking the time to review our manuscript.

Reviewer 2 Report

The article has the following areas for improvement:

1. The citation relationship between Figure.1 and Figure.2 in the main text is incorrect.

2. The English expression of the article needs to be improved, such as the difficulty in understanding the content of lines 160 to 161 of the main text (and some other areas).

3. In line 197 of the main text, it is not indicated which figure is referenced.

4. ERASOR method is an offline algorithm, and this paper applies it to online operations. What improvements have been made in real-time? What is the real-time effect?

5. Necessary annotations need to be added to Figure 4, otherwise readers will find it difficult to understand some overly simplistic expressions in the figure.

6. There is an error with the formatting of the main text in chapter 6.3.1, as the line number is missing in the main text after line 208. The same question is mentioned in chapter 2.5.

7. The article mentions ‘Region Wise Ground Plane Fitting (R-GPF)’, but lacks a corresponding explanation.

8. Please ensure the correctness of formula (14).

9. Are the analysis results of DON values based on the ULCA-MarktStreet dataset applicable to other datasets?

Author Response

(The authors gave the same response as above.)

Round 2

Reviewer 1 Report

All my comments and suggestions have been addressed.

My final comment is that the sliding window optimization displayed in Figure 5 is not properly described in the text. I suggest improving the description or deleting this figure.

Author Response

My response is in a Word file. Would you please download and review it.

Reviewer 2 Report

I have no more comment.

Author Response

Because Reviewer 1 still needs to respond, I would like to thank the referee again for taking the time to review our manuscript.